# Design and Evaluation of Short Bovine Lactoferrin-Derived Antimicrobial Peptides against Multidrug-Resistant *Enterococcus faecium*

**DOI:** 10.3390/antibiotics11081085

**Published:** 2022-08-10

**Authors:** Biswajit Mishra, LewisOscar Felix, Anindya Basu, Sai Sundeep Kollala, Yashpal Singh Chhonker, Narchonai Ganesan, Daryl J. Murry, Eleftherios Mylonakis

**Affiliations:** 1Infectious Diseases Division, Warren Alpert Medical School of Brown University, Providence, RI 02903, USA; 2School of Pharmaceutical Sciences, Rajiv Gandhi Technological University, Bhopal 462033, India; 3School of Biotechnology, Rajiv Gandhi Technological University, Bhopal 462033, India; 4Department of Pharmacy Practice and Science, College of Pharmacy, University of Nebraska Medical Center, Omaha, NE 68198, USA

**Keywords:** antimicrobial peptides, biofilm, persisters, *Enterococcus faecium*

## Abstract

*Enterococcus faecium* has become an important drug-resistant nosocomial pathogen because of widespread antibiotic abuse. We developed short and chemically simple antimicrobial peptides (AMPs) with a selective amino acid composition, fixed charge, and hydrophobicity ratio based on the core antimicrobial motif of bovine lactoferrin (LfcinB6). Among these peptides, 5L and 6L (both 12 residues long) demonstrated a narrow spectrum and high antibacterial activity against drug-resistant *E. faecium* isolates with a minimal inhibitory concentration (MIC) that ranged from 4–16 µg/mL. At 32 µg/mL, peptides 5L and 6L inhibited *E. faecium* strain C68 biofilm formation by 90% and disrupted established biofilms by 75%. At 40 µg/mL, 5L reduced 1 × 10^7^
*E. faecium* persister cells by 3 logs within 120 min of exposure, whereas 6L eliminated all persister cells within 60 min. At 0.5× MIC, 5L and 6L significantly downregulated the expression of a crucial biofilm gene *ace* by 8 folds (*p* = 0.02) and 4 folds (*p* = 0.01), respectively. At 32 µg/mL, peptides 5L and 6L both depolarized the *E. faecium* membrane, increased fluidity, and eventually ruptured the membrane. Physiologically, 5L (at 8 µg/mL) altered the tricarboxylic acid cycle, glutathione, and purine metabolism. Interestingly, in an ex vivo model of porcine skin infection, compared to no treatment, 5L (at 10× MIC) effectively eliminated all 1 × 10^6^ exponential (*p* = 0.0045) and persister *E. faecium* cells (*p* = 0.0002). In conclusion, the study outlines a roadmap for developing narrow-spectrum selective AMPs and presents peptide 5L as a potential therapeutic candidate to be explored against *E. faecium*.

## 1. Introduction

Enterococci can cause severe and difficult to treat infections, including endocarditis, bacteremia, and infections of the urinary tract [1,2]. Multidrug-resistant (MDR) enterococci are resistant to glycopeptides, β-lactams, fluoroquinolones, and aminoglycosides [3,4] and are becoming a leading cause of hospital-acquired infections [2]. In 2017, the Centers for Disease Control and Prevention (CDC) reported 54,500 infections with vancomycin-resistant enterococci (VRE) among hospitalized patients, resulting in approximately 5400 deaths in the United States [5]. Moreover, according to the CDC’s National Healthcare Safety Network 2017, *Enterococcus faecium* is the most common cause of central line-associated bloodstream infections with 70% being VRE [5,6]. The predominant *E. faecium* infections are associated with its inherent antimicrobial tolerance, biofilm-forming capabilities, and the acquisition and exchange of mobile genetic elements linked to drug-resistant genes [7].

These challenges posed by antimicrobial drug resistance, combined with sluggish antibiotic discovery demonstrate the need for innovative, efficient, and effective antimicrobial interventions [8]. Moreover, the development of agents that target enterococci requires special attention since they are part of a healthy microbiota, and a broad-spectrum antibiotic therapy can disrupt the balance of the entire microbiome [9]. On the other hand, narrow-spectrum antibacterial agents are probably less likely to promote antimicrobial resistance [10]. 

Antimicrobial peptides (AMPs) provide a promising approach for the development of novel antibacterial agents [11,12]. Small AMPs with <50 amino acid residues are part of the innate immune system found throughout all major kingdoms of life [11,13]. Most AMPs have an amphipathic alpha-helix structure with a cationic and hydrophobic surface that helps them to target bacterial membranes [12]. Interestingly, bacteria have difficulty developing resistance to AMPs, largely because modifying membranes to combat them can have adverse functional consequences [11,14]. In addition to antimicrobial properties against drug-resistant pathogens, AMPs can disrupt bacterial biofilms and kill persister cells, a physiologically quiescent subpopulation of cells tolerant to many antibiotics [12,15]. 

The toxicity in animal models or during early human studies account for 11% and 23%, respectively, of all cases of drug attrition during antimicrobial drug development [16]. Interestingly, the superior antimicrobial properties of AMPs and their low toxicity towards mammalian cells make them excellent drug candidates [12,15]. Factors such as the presence of salts may compromise the antimicrobial potency of AMPs, including beta-defensin-1, clavanins, and gramicidin S [17,18]. Recently, several broad-spectrum AMPs, including SLAY-P1 (discovered through surface-localized antimicrobial display cationic peptides) [19], Bip-P-113 (that consists of bulky non-nature amino acids) [20], and BmKn2 (derived from scorpion) [21], demonstrated excellent antimicrobial activity against *E. faecium*. However, few studies investigate the rational development of narrow-spectrum AMPs specific to MDR *E. faecium* and there is no AMP currently in the clinical trial pipeline for development as a drug [22]. 

In previous studies, others and members of our team reported the antimicrobial potential of bovine lactoferrin-derived peptides [23,24,25,26]. The pepsin digestion of the N-terminus of bovine lactoferrin results in the production of three peptides, the most active being lactoferricin B (LfcinB), which represents 17–42 amino acid residues and is effective against both Gram-positive and Gram-negative bacteria, as well as *Candida* spp. [23,24,25,26]. Interestingly, the antimicrobial core region of LfcinB (denoted as LfcinB6) contains the most miniature antimicrobial motif of lactoferrin consisting of the “RRWQWR” sequence [17,24,27].

In the present study, we expand this work and seek to design short cationic AMPs based on the smallest antimicrobial motif of LfcinB with increased hydrophobicities. We designed a series of peptides with either one or two linear copies of RRWQWR motifs with aided hydrophobic increments and selective amino acid permutations that was specifically active against *E. faecium*. To obtain peptides with the desired hydrophobicity, we either added more Leu residues to the single copy of LfcinB6 or replaced the polar glutamine with Leu first and subsequent Arg with Leu in the second stage. We investigated the mechanism of action, cytotoxic activity, killing of persister cells, and antibiofilm potential of the designed AMPs against *E. faecium*. Finally, we evaluated the efficacy of the designed AMPs in an ex vivo porcine skin infection model.

## 2. Materials and Methods

### 2.1. Bacterial Strains and Growth Conditions

Bacterial strains of *E. faecium*, *E. faecalis*, and *Staphylococcus aureus* used in this study are detailed in Appendix A. For the growth of bacteria, we used brain heart infusion (BHI) (BD, Franklin Lakes, NJ, USA) for enterococci and tryptic soy broth (TSB) (BD, Franklin Lakes, NJ, USA) for staphylococci. We used bovine lactoferrin-derived peptides with a purity of >95% (characterization data is provided in Appendix A) synthesized by solid-phase chemistry (GenScript Inc., Piscataway, NJ, USA). We used a number of *E. faecalis* and *Staphylococcus aureus* strains in order to compare the antimicrobial potency of the different lactoferrin-derived peptides.

### 2.2. Minimal Inhibitory Concentration (MIC) Assay

To determine the MICs of peptides, we used the broth microdilution method described by the Clinical and Laboratory Standards Institute [28]. We made serial dilutions of 10× concentration (10 µL) for each peptide in duplicates in 96-well plates (Cat No. 3595, Corning, New York, NY, USA). We added 90 µL of logarithmic-phase bacteria at 5 × 10^5^ CFU/mL (in BHI media) to the peptides and then incubated the plates at 37 °C for 18 h. We measured the OD_600_ absorbance readings using the Spectra Max i3x spectrophotometer (Molecular Devices, San Jose, CA, USA) and determined the MIC as the peptide concentration that inhibits the growth of bacteria.

### 2.3. MIC Assay of Designed Peptides in Various Salt Concentrations

To investigate the activity of peptides in various salt concentrations, we included physiologically relevant salts in the BHI medium. Accordingly, we added 150 mM of NaCl, 2.5 mM of CaCl_2_, 8 µM of ZnSO_4_, and 1 mM of MgSO_4_ to our peptide MIC assay.

### 2.4. Prevention of E. faecium Biofilm Formation

We evaluated the ability of the lactoferrin-derived peptides to inhibit biofilm formation following an established protocol with modifications [29]. In short, we prepared exponential cultures of *E. faecium* C68 (adjusted to OD_600_ = 0.01) in BHI media (supplemented with 0.2% glucose) from overnight grown cultures. We added 90 µL of the bacterial culture to 10 µL of serially diluted 10× peptide solution in flat-bottomed 96-well polystyrene microtiter plates (Cat No. 3595, Corning, NY, USA) and then incubated the plates at 37 °C for 24 h. We considered media containing bacteria and water as the positive control, while media with water served as the negative control. After incubation, we carefully pipetted out the media and washed the wells with PBS (Gibco, Waltham, MA, USA) to remove loosely attached planktonic cells. We quantified the biomass by staining the biofilms with crystal violet following an established protocol and quantitated the concentration of peptide that inhibits 50% of biofilm formation (MBIC_50_). Furthermore, we measured the live cell count of the biofilms using XTT [2,3-bis(2-methyloxy-4-nitro-5-sulfophenyl)-2*H*-tertazolium-5-carboxanilide] dye (ATCC, Manassas, VA, USA). We calculated the percentage of biofilm growth by assuming that there was 100% biofilm growth in wells with bacteria without peptide treatment.

### 2.5. Disruption of E. faecium Established Biofilms

We prepared exponential cultures of *E. faecium* strain C68 (adjusted to 1 × 10^8^ CFU/mL) in BHI media (supplemented with 0.2% glucose) from overnight grown cultures. We added 100 µL of the bacterial culture to wells of flat-bottomed 96-well polystyrene microtiter plates (Cat No. 3595, Corning, NY, USA). We then incubated the plates at 37 °C for 24 h in static conditions to allow biofilm formation. After incubation, we carefully pipetted out the media and washed the wells with PBS (Gibco, Waltham, MA, USA) to remove loosely attached planktonic cells. We treated the established biofilms in each well with 10 µL of serially diluted 10× peptide solution, followed by the addition of 90 μL of BHI media (supplemented with 0.2% glucose). The plates were incubated for another 24 h at 37 °C before processing for biomass and live-cell contents. We quantitated the concentration of peptide that disrupts 50% of established biofilms (MBEC_50_).

### 2.6. Confocal Microscopy of Peptide-Treated E. faecium Established Biofilms 

We prepared exponential cultures of *E. faecium* strain C68 (adjusted to 1 × 10^6^ CFU/mL) in BHI media (supplemented with 0.2% glucose) from overnight grown cultures. To establish biofilms, we added 1 mL of the bacterial culture to the chambers of cuvettes (borosilicate cover glass, Nunc Cat. No. 155380) and maintained them at 37 °C for 24 h under static conditions [30]. We carefully removed the media from the wells after incubation and then washed the wells with PBS (Gibco, MD, USA) to remove loosely attached planktonic cells. We treated the established biofilms in each well with 100 µL of serially diluted 10× MIC peptide solution (5L and 6L) followed by the addition of 900 μL of BHI media (supplemented with 0.2% glucose) and incubated the cuvettes for an additional 24 h at 37 °C. After incubation, we washed the cuvettes carefully with PBS (Gibco, MD, USA) to remove the planktonic cells and stained the biofilms with 10 µL of LIVE/DEAD kit (Molecular Probes, Life Technologies, Eugene, OR, USA) according to the manufacturer’s instructions. Finally, we visualized the biofilms using a confocal microscope (Zeiss 880) and processed the data via Zen 2010 software (Carl Zeiss, Jena, Germany).

### 2.7. Quantitative Polymerase Chain Reaction (qPCR) 

For the qPCR of genes associated with biofilm formation, we followed an established protocol for RNA isolation and quantitative PCR reactions with minor modifications [15]. We prepared exponential cultures of *E. faecium* strain C68 adjusted to an OD_600_ = 0.4 in BHI media from overnight grown cultures. After treating 10 mL of bacterial culture with 0.5× MICs of 5L and 6L for 30 min, we extracted RNA using the RNeasy mini kit (Qiagen, Hilden, Germany) according to the manufacturer’s instructions. Using the primers listed in Appendix A, we performed cDNA synthesis and quantitative reverse transcription (RT)-PCR as recommended by the manufacturer (Bio-Rad, Hercules, CA, USA). Our qPCR cycling conditions were: 95 °C for 30 s; 40 cycles at 95 °C for 5 s; 55 °C for 30 s; and finished with a melt curve analysis from 65 to 95 °C.

### 2.8. E. faecium Persister Cell and Time-Kill Assays

For the generation of antibiotic-induced persister cells, we followed an established protocol [31]. In brief, 25 mL of *E. faecium* strain C68 culture was grown to the stationary phase and then treated with gentamicin at 400 μg/mL for 5 h. We then disposed of the antibiotic-containing media and regrew the bacteria for another 18 h in fresh media. To produce stable persister cells, we repeated the cycle of antibiotic exposure and regrowth five times. The bacteria culture was then washed with the same volume of phosphate-buffered saline (PBS) 3× times and adjusted to 1 × 10^7^ CFU/mL in PBS. For killing kinetics of the persister cells, we added 1 mL of the cell suspension to the wells of a 2 mL deep well assay block (Cat No. 3960, Corning, NY, USA) containing 10× MICs of 5L and 6L and incubated them at 37 °C, with shaking at 225 rpm. At specific times (0, 15, 30, 60, and 120 min), we collected 20 μL samples, diluted them serially, and then plated them on tryptic soy agar (TSA) (BD Difco, Franklin Lakes, NJ, USA) plates for 18 h at 37 °C for colony counting. We performed the experiments in duplicate. As a comparison, we prepared the exponential cells of *E. faecium* strain C68 in PBS and analyzed their killing kinetics in the same manner as described for persister cells. 

### 2.9. Persister Cell Membrane Permeability Assay

We generated *E. faecium* strain C68 persister cells (described in the above section) and diluted them in PBS to OD_600_ = 0.2. We added SYTOX Green (Molecular Probes, Life Technologies, Eugene, OR, USA) to the diluted persister suspension to a final concentration of 5 μM and incubated them for 30 min at room temperature in the dark. We added 90 µL of the dye/bacteria mixture to 10 μL of serially diluted 10× peptide concentration in a 96-well black, clear-bottom plate (Cat No. 3904, Corning, NY, USA) and recorded fluorescence on a SpectraMax i3x (Molecular Devices, San Jose, CA, USA) with excitation and emission wavelengths of 485 nm and 525 nm, respectively, for 1 h at room temperature. For comparison, we also conducted membrane permeability assays using exponential cells and followed similar protocol as described for persister cells.

### 2.10. Membrane Depolarization

We measured bacterial membrane potential based on an established protocol with minor modifications [15]. We prepared an exponential phase culture of *E. faecium* strain C68 in fresh BHI media, washed it twice in PBS, and resuspended it in a double volume of PBS. To energize the bacterial cells, we added 25 mM of glucose for 15 min at 37 °C, followed by 500 nM of DiBAC4(3) (bis-(1,3-dibutylbarbituric acid)trimethine oxonol) (Molecular Probes/Thermo Fisher Scientific, Mississauga, ON, Canada). The DiBAC4(3) is a potential sensitive probe that enters depolarized cells, binds to intracellular proteins or membranes, and exhibits enhanced fluorescence. We distributed 90 µL of this bacterial culture in a 96-well black plate with a clear bottom (Cat No. 3904, Corning, NY, USA) and monitored the fluorescence for 20 min at excitation 485 nm and emission 520 nm, respectively, until the baseline was stable. Then, we added 10 µL of serially diluted peptide solutions and recorded the fluorescence for another 40 min. We included Triton X-100 (1%) as a positive control. 

### 2.11. Laurdan-Based Membrane Fluidity Assay

We measured the peptide-induced changes to the bacterial membrane fluidity following an established protocol [15]. Briefly, we washed the exponential phase of *E. faecium* strain C68 bacteria three times with PBS and resuspended in half the volume of the initial culture taken for washing. To the bacterial suspensions in PBS, we added the Laurdan dye (Cat No. 40227, Sigma-Aldrich, Darmstadt, Germany) at a final concentration of 10 µM at room temperature in the dark and distributed 100 µL of this dye/bacteria mixture into 96-well transparent black plates (Cat No. 3904, Corning, New York, NY, USA) containing a 10× peptide solution. After incubation at room temperature for one hour in the dark, we measured the fluorescence intensity using a SpectraMax i3x (Molecular Devices, CA, USA), with excitation at 350 nm and dual emissions at 435 nm and 490 nm. Accordingly, we calculated the Laurdan generalized polarization (GP) using the formula GP = (I_435_ − I_490_)/(I_435_ + I_490_). Our positive control included 50 mM of benzyl alcohol as a membrane fluidizer.

### 2.12. Propidium Iodide-Based Membrane Permeability

We used a fluorescence-based bacterial permeation assay to establish membrane-oriented interactions of the designed peptides. Briefly, we washed the exponential phase of *E. faecium* strain C68 bacteria three times with PBS and adjusted it to OD_600_ = 0.4 in PBS. We added PI dye to the bacteria to achieve a 2 µM final dye concentration and distributed 90 μL of this bacteria/dye mixture into a 96-well black clear-bottom plate (Cat no. 3904, Corning, New York, NY, USA) containing 10 μL of serially diluted 10× peptide concentrations. Subsequently, we monitored the plate fluorescence on a SpectraMax i3x (Molecular Devices, San Jose, CA, USA) with an excitation wavelength of 584 nm and an emission wavelength of 620 nm for 1 h at room temperature.

### 2.13. Growth Inhibition Assay

We cultivated overnight cultures of *E. faecium* strain C68 in BHI media. We inoculated the bacteria in a fresh BHI medium the next day to achieve the exponential phase and adjusted the bacterial density to 1 × 10^8^ CFU/mL. We added 90 µL of this bacterial culture to each well of the microtiter plates (Cat No. 3595, Corning, New York, NY, USA) containing 10 µL of serially diluted 10× peptide solution. We monitored the bacterial growth on a SpectraMax i3x spectrophotometer (Molecular Devices, San Jose, CA, USA) by taking OD_600_ readings every 10 min for 4 h. 

### 2.14. ATP Release Assay

To determine the ATP leakage potential of the peptides, we employed a luciferase-based assay detailed in [32]. In brief, we washed the exponential phase of *E. faecium* strain C68 bacteria three times with PBS (pH 7.4) and adjusted it to OD_600_ = 0.4 in PBS. Following treatment of the bacteria with 32 µg/mL of 5L and 6L peptides at 37 °C for 30 min, we centrifuged the cells at 14,000 rpm for 5 min. We transferred 50 µL of the supernatant of each sample to each well of black, clear-bottom, 96-well plates (Cat No. 3904, Corning, New York, NY, USA) containing 50 µL of BacTiter-Glo reagent (Promega, Madison, WI, USA) and measured plate luminescence after 5 min of incubation at room temperature in the dark with a SpectraMax i3x reader (Molecular Devices, San Jose, CA, USA).

### 2.15. Molecular Dynamics Simulation

We performed molecular dynamics (MD) simulations in the presence of POPE: cardiolipin (3:1) membranes with 128 molecules of the desired lipid system. We used the lipid composition of *E. faecium* that was determined previously using biochemical lipid extraction methods [33]. We constructed the starting structure of the peptide 5L with Alphafold2 server [34]. We placed the peptide 5L at least 1.5 nm away from the membrane system above the upper leaflet in parallel to the upper membrane leaflet at the beginning of the MD run. The initial system configuration was generated using the Charmm-GUI program [35]. We initially simulated the system for at least 100 ns using the Martini 22p coarse-grain forcefield following which the system was converted into an all-atom configuration using the Charmm-GUI program [36,37] and further simulated for 1 ns using the Charm-36 forcefield. We performed all of the simulations using the GROMACS version 2020.1-1 software package. In all of the simulations, we maintained a temperature of 310 K and hydrated the system with a water (TIP3P) layer thickness of 22.5 Å. For visualization and subsequent analysis of the simulation data, we used the VMD and PyMOL software packages.

### 2.16. Metabolomics Analysis

For the metabolomics sample preparation, we used exponential cultures of *E. faecium* strain C68 grown in BHI media adjusted to OD_600_ = 0.5. We treated 10 mL of this bacterial culture with 2× MIC of peptide 5L or water (for bacterial control) for 30 min at 37 °C (*n* = 3, technical replicates). After that, we washed the bacteria 3× times with PBS and stored the pellet at −20 °C. At a later stage, to each cell pellet, we added 0.2 mL of 2-propanol: 100 mM of NH_4_HCO_3_, pH 7.4 (1:1 *v*/*v*), and sonicated the cells. For spiking, we added 20 µL of IS to each sample and vortexed them. Subsequently, we added 1.0 mL of methanol for deproteinization and cooled the mixture at −20 °C for 10 min. After centrifugation for 10 min at 14,000× *g* at 4 °C, we transferred the supernatants into glass tubes and dried them under a stream of nitrogen at 40 °C. Finally, we dissolved the residues in 100 µL of water and injected 3 µL of this solution into an LC-MS/MS 8060 system (Shimadzu Scientific Inc., Columbia, MD, USA), equipped with a DUIS source operated in both positive and negative electrospray ionization modes. For the liquid chromatographic analysis, we used the Nexera UPLC system (Shimadzu Scientific Inc., Columbia, MD, USA). We quantitated primary metabolites (~150) following an established LC-MS/MS method [38].

### 2.17. Hemolysis of Human Red Blood Cells (hRBCs)

We evaluated the ability of the peptide to cause hemoglobin leakage using the method described in [15]. In brief, we washed human erythrocytes (Rockland Immunochemicals (Limerick, Pottstown, PA, USA)) 3× times in an equal volume of PBS and resuspended them as 2% hRBCs solution. We added 90 µL of the blood cells to 10 µL of the peptide solution (10×) in a 96-well microtiter plate (Cat No. 3595, Corning, New York, NY, USA) and incubated the plates at 37 °C for 1 h. Next, we centrifuged the plates at 500× *g* for 5 min, transferred 60 µL of the supernatant into a fresh 96-well plate, and read the absorbance at 540 nm. We calculated the percent hemolysis considering 100% hemolysis caused by 1% Triton X-100, and 0% on PBS and using the formula for percentage of hemolysis as: (A_540_ nm in the peptide solution − A_540_ nm in PBS)/(A_540_ nm of 1% Triton X-100-treated sample − A_540_ nm in PBS) × 100.

### 2.18. Mammalian Cell Cytotoxicity Assays

We used liver-derived HepG2 cells to evaluate AMP cytotoxicity behavior in mammalian cells based on an established protocol [15]. We maintained the HepG2 cells at 37 °C in 5% CO_2_ in Dulbecco’s Modified Eagle Medium (DMEM) (Gibco, Waltham, MA, USA) supplemented with 10% fetal bovine serum (FBS) (Gibco, Gaithersburg, MD, USA) and 1% penicillin/streptomycin (Gibco, Gaithersburg, MD, USA). Next, we harvested and resuspended the cells in a fresh medium and distributed 50 µL (1 × 10^6^ cells) in a 96-well plate containing 50 µL of serially diluted AMPs in serum and antibiotic-free DMEM and incubated the plates at 37 °C in 5% CO_2_ for 24 h. Before the end of the incubation period (at the 20th hour), we added 10 µL of 2-(4-iodophenyl)-3-(4-nitrophenyl)-5-(2,4-disulfophenyl)-2*H*-tetrazolium (WST-1) (Roche, Mannheim, Germany) to each well and monitored the reduction of WST-1 at 450 nm using a SpectraMax i3x (Molecular Devices, San Jose, CA, USA). We performed the assays in triplicates and calculated the percentage of cell survival.

### 2.19. Ex Vivo Porcine Skin Infection Assay

We evaluated the efficacy of designed lactoferrin AMPs in an ex vivo porcine skin model previously described with minor modifications [39]. We first cleaned the shaved porcine skin (Fisher Scientific, Hampton, NH, USA) with 70% ethanol and then water, and glued polyethylene tubing (5 mm) (Nalgene VWR 228-0170, Radnor, PA, USA) to the skin samples with cyanoacrylate glue (Fisher Scientific, Hampton, NH, USA). Using disposable razors, we cut each piece of skin glued to the tube and inserted it into a 12-well tissue culture plate (Falcon, Cat No. 353182). We infected each skin with 1 × 10^6^ CFU of exponential or persister *E. faecium* strain C68 cells in a total volume of 10 µL. Following an incubation of 1 h at 37 °C, we treated the skin with 50 µL with 5L and 6L at concentrations of 10× or 1× MIC. After an additional 4 h of incubation at 37 °C, we washed each tubing well with 250 µL of PBS (pH 7.4) containing 0.05% Triton X-100 to remove bacteria. After serially diluting the bacterial sample, we counted their CFUs.

### 2.20. Statistical Analysis

We used one way ANOVA for the statistical analysis in all of the experiments except for real-time PCR and metabolomics where we employed Student’s *t*-test. For all of the experiments, we considered *p* < 0.05 as significant.

## 3. Results

### 3.1. Design and Bactericidal Activity of Short LfcinB6-Derived Peptides

To design simple and short AMPs for further antibacterial agent development, we considered peptides with only one or two copies of the LfcinB6 core antimicrobial motif. The native LfcinB6 peptide has a net charge of +3 and a hydrophobic content of 33%. Despite the three Arg residues that can bind effectively to bacterial membranes, the peptide is inactive against *E. faecium*. Therefore, we aimed to improve the hydrophobicity of the LfcinB6 peptides.

We used a single copy of LfcinB6 to design AMPs with increased hydrophobic contents of ~60% (Table 1). To increase the overall hydrophobic contents in LfcinB6, we either added more Leu residues or replaced the polar, uncharged amino acids. Subsequently, we added a cluster of four consecutive Leu residues to LfcinB6 C-terminus in peptide 1L to reach a hydrophobicity of 60% or substituted the non-hydrophilic Gln4 residue with Leu and then added a Leu residue to the C-terminus in peptide 2L to reach a hydrophobicity of 57%. With increased length (10 amino acid residues) and hydrophobicity, peptide 1L showed an MIC of 16 µg/mL against MDR *E. faecium* strain C68. 

The short peptide 2L (seven amino acid residues) was ineffective (tested up to 32 µg/mL), so we decided to use two copies of LfcinB6 for the next round of peptide design. Two copies of LfcinB6 have 12 amino acid residues imparting 33% hydrophobicity, and in order to maintain 58% hydrophobicity, three hydrophobic substitutions are required. As a result, we replaced both Gln4/10 with Leu, followed by a sequential Arg to Leu substitutions (at positions 1,2,6,7,8, and 12) in successive peptides. Peptide 3L with arg1 replaced with Leu had an MIC of 16 µg/mL. Other AMPs (4L–8L) with Arg to Leu substitution showed excellent antimicrobial activity against *E. faecium* strain C68 with MIC values of 4 µg/mL. The designed lactoferrin peptides did not show antibacterial activity toward *S. aureus* and *E. faecalis*, except peptide 6L with a very high MIC of 32 µg/mL against *S. aureus* MW2 (Appendix A). 

Next, to evaluate the robustness of the antibacterial activity of the designed LfcinB6 peptides, we included physiological salts into the MIC assays. In the presence of 150 mM of NaCl, 8 mM of ZnSO_4_, and 1 mM of MgSO_4_, the MIC of peptides 1L–8L (except 2L) was influenced by only 2-fold (Table 2). However, in the presence of 2.5 mM of CaCl_2_, peptides 1L, 3L, and 4L lost their antibacterial activity (MIC ≥ 32 µg/mL) while the MICs of peptides 5L–8L were only increased by 2–4 folds. 

To further evaluate the antimicrobial potency of the LfcinB6 peptides, we tested the activity of peptides 5L–8L against drug-resistant clinical isolates and other bacterial physiological forms, such as biofilms and antibiotic-induced persister cells. The MICs of peptides 5L–7L against clinical isolates of *E. faecium* ranged from 4–16 µg/mL, while that of peptide 8L ranged from 8–32 µg/mL against vancomycin-sensitive strains D14, D24, D25, and D29 and vancomycin-resistant *E. faecium* strains C68 and WC176 (VRE) (Table 3). However, all peptides 5L–8L showed an MIC of 4 µg/mL to a tetracycline-resistant *E. faecium* strain (E007).

### 3.2. Antibiofilm Effects of LfcinB6-Derived Peptides

Next, we tested the ability of peptides 5L–8L to inhibit the biofilm formation and disrupt established biofilms of *E. faecium* strain C68 (Figure 1A–D). Peptides 6L (at 1.2 µg/mL) and 8L (at 2.8 µg/mL) killed 50% of live bacterial cells during the biofilm formation stage, while peptide 5L (MBIC_50_ 1 µg/mL), followed by 8L (MBIC_50_ 2 µg/mL) reduced the biomass contents (Figure 1B).

During the biofilm formation stage, at the highest tested concentration (32 µg/mL), 6L depleted 92% of the biomass, followed by 5L (90%), 7L (85%), and 8L (80%). In the more robust condition of 24 h established biofilms, the peptide 5L killed 50% of the live bacteria contained within the matured biofilm matrix (Figure 1C). Peptides 5L and 6L were most potent and eradicated the established biomass with MBEC_50_ values of 16 µg/mL (Figure 1D). Peptide 8L had an MBEC_50_ of 30 µg/mL, whereas peptide 7L failed to achieve an MBEC_50_ within the concentration range tested. At 32 µg/mL, 5L and 6L were equally effective peptides and reduced 75% of the established biomass. 

Additionally, to verify the robust antibiofilm properties of peptides 5L and 6L, we examined their ability to disrupt 24 h established biofilms of *E. faecium* E007 (a tetracycline-resistant strain) (Appendix A). Indeed, peptides 5L and 6L were effective on biofilms. Peptides 5L and 6L killed 50% of the live bacteria cells at 8 µg/mL, but 7L and 8L failed to kill 50% even at 32 µg/mL. At 32 µg/mL, peptides 5L (MBEC_50_ 10 µg/mL) and 6L (MBEC_50_ 8 µg/mL) disrupted 75% of the established biomass, while peptides 7L and 8L disrupted only 25–30%. 

Further by employing confocal laser scanning microscopy, we confirmed that both peptides 5L and 6L (at 10× MIC) killed most of the bacteria inside the biofilms. Untreated *E. faecium* strain C68 biofilms (Figure 1E) were green in color, representing live cells, whereas peptide-treated biofilms (Figure 1F,G) are dominated by red color, which indicates dead cells. In general, we found that 5L and 6L had the most potent antibiofilm abilities of the LfcinB6 peptides. 

In addition, to uncover the mode of antibiofilm activities of peptide 5L and 6L at the genetic level, we measured the relative expression of significant biofilm-regulating genes in *E. faecium* strain C68. The peptides 5L and 6L at subinhibitory concentrations (0.5× MIC) suppressed the expression of crucial biofilm genes (Figure 1H). Peptide 5L significantly downregulated the expression of *ace* (8-fold), *esp* (4-fold), *ebpA* (2-fold), and *ebpC* (2-fold). Whereas 6L significantly downregulated the expression of *ace* (4-fold) and *ebpC* (2-fold).

### 3.3. Antipersister Effects of LfcinB6 Peptides

To further demonstrate the merits of peptides 5L and 6L, we evaluated their antibacterial potential against persister cells, which are tolerant to many antibiotics. At 10× MIC, peptide 5L reduced the viable 10^7^ CFU of a culture with *E. faecium* strain C68 persister cells by 3 logs, while peptide 6L was even more active (Figure 1I).

A SYTOX-based fluorescent assay also confirmed the rapid permeabilization of persister cell membranes. In this series of experiments, we observed a time-dependent fluorescent increment upon exposure of 16 µg/mL of peptides 5L and 6L to a mixture of STYOX dye and persister cells. Untreated bacteria as well as the ampicillin control failed to increase fluorescence (Figure 1J).

Further, at 10× MIC, both peptides 5L and 6L eliminated all 10^7^ CFU of exponential cells of *E. faecium* strain C68 after 120 min and 60 min, respectively (Figure 1K). As evidenced by increased fluorescent readings, the peptides also displayed enhanced membrane permeabilization in the SYTOX assay against exponential cell membranes treated at 16 µg/mL (Figure 1L).

### 3.4. Mechanism of Action (MOA) of LfcinB6 Peptides

In order to study the MOA of LfcinB6-derived peptides, we conducted peptide-membrane interaction studies. First, we employed a fluorescence-based transmembrane potential measurement assay that uses the DIBAC_4_(3) dye (described in Methods). *E. faecium* strain C68 cell membrane exposure to peptides 5L–8L (at 16 µg/mL) induced a rapid fluorescence rise, indicating change in transmembrane potential (Figure 2A). Next, we measured bacterial membrane fluidity by using a Laurdan dye (Figure 2B). At 32 µg/mL, the generalized polarization (GP) values of the peptides 5L–8L were comparable to those of 50 mM of benzoyl alcohol (a known membrane fluidizer), indicating a significant decrease in membrane fluidity. 

We also employed fluorescence-based dye permeation experiments to understand the membrane interactive nature of the peptides. The membrane-impervious dye, PI, rapidly penetrated the bacterial cells in the presence of peptides 5L–8L at 32 µg/mL (Figure 2C). Peptide 6L, followed by 5L, increased fluorescence immediately, while 7L and 8L developed at a slower pace. The antibiotics vancomycin and ampicillin (that do not target the membrane), and negative controls, did not increase fluorescence. Following the PI penetration trends, peptide 6L inhibited the maximum growth of *E. faecium*, followed by peptides 5L, 7L, and 8L (Figure 2D). 

Furthermore, we measured the ATP leakage from *E. faecium* strain C68 in the presence of LfcinB6 peptides. *E. faecium* strain C68 cells exposed to 32 µg/mL of peptides 5L or 6L displayed increased luminescence levels, indicative of ATP leakage (Figure 2E). 

### 3.5. MD Simulation of Peptide 5L in E. faecium Membrane Mimetic Model

To obtain structural insights to key amino acids residue interactions of the peptide 5L with the bacterial membrane, we employed MD simulations. We performed the simulation of peptide 5L in the presence of an *E. faecium*-mimicking membrane system. Intriguingly, the peptide bound with the membrane system within 15–25 ns (Figure 2F) and remained bound to the upper leaflet of the membrane for the rest of the 100 ns simulation. This is also corroborated by the partial density plot of the system, which suggests that the peptide predominantly remained about 1.25 nm inside the upper membrane leaflet (Appendix A). 

Peptide 5L adopted a non-helical amphipathic conformation (Figure 2G) with several stable hydrogen bonds to the nearest POPC and cardiolipin molecules (Appendix A). As shown in Figure 2G, except for Leu6, all of the other hydrophobic residues were present in the hydrophobic face (with Leu10 predominantly interacting with the cardiolipins), separated from Arg residues in the hydrophilic front. Especially for the latter case, we observed that Arg-8, 7, and 1 primarily interacted with the POPE lipids. In summary, the membrane interaction studies followed by molecular simulations suggested a membrane-interactive MOA of the LfcinB6 peptides.

### 3.6. Metabolomic Changes in E. faecium Physiology in Presence of LfcinB6 Peptide

In order to study for any effects in the bacterial physiology, we measured the change in primary metabolite levels of *E. faecium* in the presence of peptide 5L. We employed a LC-MS/MS targeted metabolomics on *E. faecium* strain C68 cells treated with 5L peptide (at 2× MIC). In the principal component analysis (PCA) and partial least squares-discriminant analysis (PLSDA) plots, we found distinct metabolic alterations between the *E. faecium* control and 5L-treated conditions with three technical replicates (Appendix A). Interestingly, 13 primary metabolites were significantly altered in 5L-treated groups with a *p* value cutoff of 0.05 (Appendix A). Peptide 5L downregulated serine the most (0.03-fold) and upregulated putrescine (7-fold) (Appendix A). A pathway impact analysis showed that 5L caused significant changes in *E. faecium* metabolites involved in the tricarboxylic acid cycle, glutathione, cysteine, methionine, and purine metabolism (Figure 2G).

### 3.7. Cytotoxicity Analysis of LfcinB6 Peptides

In the next series of experiments, we evaluated the toxicity of LfcinB6 peptides against mammalian cells, including human red blood cells and human liver-derived HepG2 cell lines. Peptide 5L had an HL_50_ of 56 µg/mL, whereas peptide 6L had an HL_50_ of 34 µg/mL (Figure 3A). In contrast to red blood cells, neither peptides 5L nor 6L caused cell death in HepG2 cells up to a concentration of 128 µg/mL (Figure 3B).

### 3.8. Ex Vivo Antibacterial Efficacy of LfcinB6-Derived Peptides

Finally, to establish the antimicrobial potential of LfcinB6-derived peptides in a higher mammalian model, we examined the potency of peptides 5L and 6L in an ex vivo porcine skin model (Figure 4). At 10× MIC concentrations, peptides 5L and 6L effectively eliminated almost all exponential cells (Figure 4A). At 1× MIC, peptides 5L and 6L only reduced the bacterial load by 1 log and 1.2 logs, respectively. Gentamicin was used as a control and was ineffective in reducing the bacterial burden. We found similar trends of bacteria eradication by peptides 5L and 6L against *E. faecium* strain C68 persister cells (Figure 4B). A concentration of 10× MIC of 5L and 6L completely eliminated persister cells, whereas treatment at 1× MIC resulted in only a 1 log reduction in the bacterial load.

## 4. Discussion

*E. faecium* is associated with an increasing number of resistant, severe, and difficult to treat infections [5,6]. AMPs may provide interesting insights in antimicrobial drug discovery due to their mode of action, host cell selectivity, and ability to modulate the immune system [11,12,13,14,41]. In the present study, we developed short peptides based on the smallest antimicrobial core region of bovine lactoferrin (LfcinB6) that is specifically active against *E. faecium*. Among these peptides, 5L and 6L killed antibiotic-induced *E. faecium* persister cells in vitro and in an ex vivo porcine skin infection model. These peptides demonstrated a membrane-targeting mechanism of action and antibiofilm activity by significantly downregulating *ace* and *ebp* genes responsible for biofilm adherence. 

The high cost of peptide production is a hinderance in the clinical and commercial development of AMPs [42]. To overcome this issue and to make the process of drug development simpler, researchers have used the smallest antimicrobial motifs of many AMPs, such as KR12 of human LL-37 [18], G-WKRKRF-G of temporins [43], GXXXG in glycophorin A [44], and RRWQWR in LfcinB6 [25,27,45], to design short and potent AMPs. In this study, we developed narrow-spectrum, linear, and short AMPs against *E. faecium* and selected the peptide LfcinB6 because of its short antimicrobial core region, simple amino acid composition, and as it is a non-toxic part of human diet [25,26]. Moreover, previous reports on LfcinB6 focused on its antimicrobial activity against common pathogens, including *S. aureus*, *Klebsiella pneumoniae*, *Pseudomonas aeruginosa*, and *Escherichia coli* [17,27,46]. However, previous studies on the activity of lactoferrin-derived peptides against *E. faecium* are limited to a chemically complex branched dimeric or palindromic LfcinB6 peptide with high hydrophobicity [27]. 

A previous physicochemical analysis of 138 AMPs active against *E. faecium* revealed an average net charge of +3.87 and a hydrophobicity of 58.2% [47]. Thus, in this study we decided to increase the hydrophobicity of linear LfcinB6 peptides to increase their antimicrobial potency against *E. faecium.* It is important to note that the potent antibacterial activity of the peptides designed with two copies of LfcinB6 (such as peptides 4L–8L) agrees with our previous observations on cecropin-4-derived peptides, where we found that a minimum amino acid length is required in AMPs to maintain their antimicrobial activity [15]. Moreover, the narrow activity spectrum of peptides 4L–8L is consistent with that of DFTamP1 [41] and APO-type designed peptides [48], each with a unique amino acid composition and a specific charge to hydrophobicity balance.

Salt decreases the potency of AMPs due to charge screening effects as both AMPs and salts compete for bacterial membrane binding [49]. For example, major AMP families, including beta-defensin-1, clavanins, gramicidin S, P-113 (a fragment of Histatin 5), and linear and tetrameric LfcinB6, are salt sensitive [17,18]. In this context, AMPs resistant to salt are desirable for developing peptide antibiotics against *E. faecium*. Salt-resistant linear peptides such as LfcinB6-KR-12 hybrid peptide, cecropin-4-derived C18, and arginine-rich decamer peptides D5 and D6 have high hydrophobicity and charges like LfcinB6-designed peptides, suggesting salt sensitivity in AMPs is template-dependent [15,18,49]. High cationic peptides are predicted to neutralize the charge screening effect induced by the addition of salt [18]. Interestingly, our results showed that even in the presence of various salts, the antibacterial properties of peptides 5L–8L were minimally influenced (Table 2). In peptides 5L–8L, the five Agr residues effectively neutralized the charge screening effects of monovalent Na^+^ and divalent Zn^2+^, Mg^2+^, and Ca^2+^ cations.

Bacterial cells in biofilms and persister states are involved in chronic and relapsing infections that are difficult to eradicate with conventional antibiotics [50,51]. To our knowledge, this is the first report describing linear peptides derived from LfcinB6 that have antibiofilm and antipersister properties against *E. faecium*. Notably, peptides 5L and 6L did not only kill persister cells of *E. faecium* but also inhibited the biofilm formation process and disrupted 24 h established biofilms (Figure 1). Crystal violet and XTT staining of peptide-treated *E. faecium* biofilms showed significantly reduced biomass contents and fewer live cells. In addition, confocal microscopic imaging using a live–dead staining confirmed predominant red-colored dead cells and low-density biofilms in 5L- and 6L-treated samples compared to the control. Bovine lactoferrin-derived peptides showed promising antibiofilm potential against *P. fluorescens* [52], Listeria isolates and *E. coli* [53], *A. baumannii* [54], and *P. aeruginosa* [55]. Interestingly, our observation of peptide-treated samples with reduced biofilm thickness and density with confocal microscopy concurs with a study by Ajish et al., which found that the Lf-KR-12 peptide weakens the biofilm matrix by forming pores within lipid components [18].

Our study supports the finding that, sub-MIC concentrations of antibiotics inhibit essential biofilm genes functions, thereby affecting bacterial adhesion to abiotic substrates [56]. The use of subinhibitory concentrations of antibiotics is one of several methods of gene targeting to affect overall virulence [57]. Interestingly, peptide 5L (at sub-MIC) downregulated the key biofilm genes of *E. faecium*, including *ace*, *esp*, and *ebp*, and thus resulted in biofilm inhibitions and disruption. These genes are associated with *E. faecium* pathogenesis and biofilm formation [58]. More specifically, *ace* facilitates colonization by interacting with human proteins such as collagen type I, IV, laminin, and dentin [59], while *esp* and *ebp* play a role in the initial adherence of bacteria to a substrate during biofilm formation and contribute to biofilm-associated infections, including experimental endocarditis and urinary tract infections [60]. 

LfcinB6 is amphipathic with Arg residues present on the hydrophilic face, and Trp residues making up the hydrophobic face [61]. The LfcinB6 peptides we designed were also amphipathic (Appendix A). During the initial interaction, peptides 5L–8L depolarized the *E. faecium* cell membrane, which was followed by a change in fluidity and membrane disruption (Figure 2). Cationic AMPs exert their antimicrobial action via targeting the bacterial membrane [12]. Substituting Arg residues with Leu residues in the LfcinB6-designed peptides increased the peptides’ hydrophobicity. Cationic peptides designed from KR-12 (Arg substituted with Ala) [62], Tetra-F2W (Agr substituted with Trp) [63], and cecropin-4 peptides (Gly substituted with Leu) [15] showed increased antimicrobial potency when hydrophobic amino acids substituted a charge or neutral amino acid. Likewise, replacing the less hydrophobic amino acid residues with a higher hydrophobic amino acid in peptides such as L10 (Ala replaced with Trp) [23], DFTamp1 (Val or Iso replaced with Leu) [41], and C18G (Val replaced with Leu) [64] also increased their antimicrobial potencies. Interestingly, a human β-defensin-28 variant D6 peptide with more Trp residues binds 1-Palmitoyl-2-oleoyl-sn-glycero-3-(phospho-rac-(1-glycerol)) (POPG, a bacterial membrane mimetic vesicle) more efficiently than its less Trp counterparts, with a 10^4^ order of magnitude difference in the binding constant [49]. In agreement, the LfcinB6-derived cationic peptides reported here bind to bacterial membranes initially through electrostatic interactions, followed by interactions between the hydrophobic tryptophan residues and the lipid bilayer resulting in membrane rupture. Molecular dynamics simulations also suggested the amphipathic orientation of peptide 5L on the membrane surface with Arg on the hydrophilic face and trp and Leu on the hydrophobic face (except Leu6). Peptide 5L penetrated an *E. faecium* mimetic membrane (3:1 POPE: cardiolipin) model within 25 ns. Interestingly, a membrane-active lipopeptide of lactoferrin, acyl-LfcinB6, also binds to 3:1 POPE: POPG membrane within 75 ns, thus, confirming the membrane-targeting mechanism of LfcinB6 peptides [65]. 

In addition to targeting the bacterial membrane, LfcinB peptides also affect ribosomes, mitochondria, and other physiological functions [66,67]. We performed targeted metabolomics to identify peptide-altered bacterial metabolic pathways. Based on enrichment analysis, we observed that the tricarboxylic acid cycle, glutathione, and purine metabolism were the top enriched metabolic pathway in 5L-treated *E. faecium* (Appendix A). Antibacterial compounds affecting energy dissipation and the tricarboxylic acid cycle lowers cellular energy levels by accelerating glutamate oxidative decarboxylation which produces more NADH^+^ and ketoglutarate acid to be used in oxidation [68]. Additionally, *E. coli* reduces periplasmic glutathione to resist many antibiotics, including trimethoprim (DNA targeting) [69]. Furthermore, purine metabolism is an effective antimicrobial drug target due to its role in nucleic acid synthesis [70]. AMPs such as Bactenecin 7 and LfcinB are known to alter purine metabolism [70]. Based on these observations, the LfcinB6-derived peptide 5L may be capable of exerting multiple mechanisms of action in addition to membrane targeting. 

## 5. Conclusions

In summary, we developed short LfcinB6 peptides with activity against *E. faecium*, including persister cells and biofilms. The design strategy adopted here with charge and hydrophobicity balance could serve as a roadmap for developing AMPs that are bacterial selective. The lead peptide, 5L, exhibited narrow-spectrum activity, high antimicrobial potency, low mammalian cell cytotoxicity, and membrane-activating mechanisms, making it an attractive therapeutic candidate. Transcription-based mechanism assays and in vivo testing in higher organisms will be needed before 5L can move on to the next phase of anti-infective drug development.

## Figures and Tables

**Figure 1 antibiotics-11-01085-f001:**
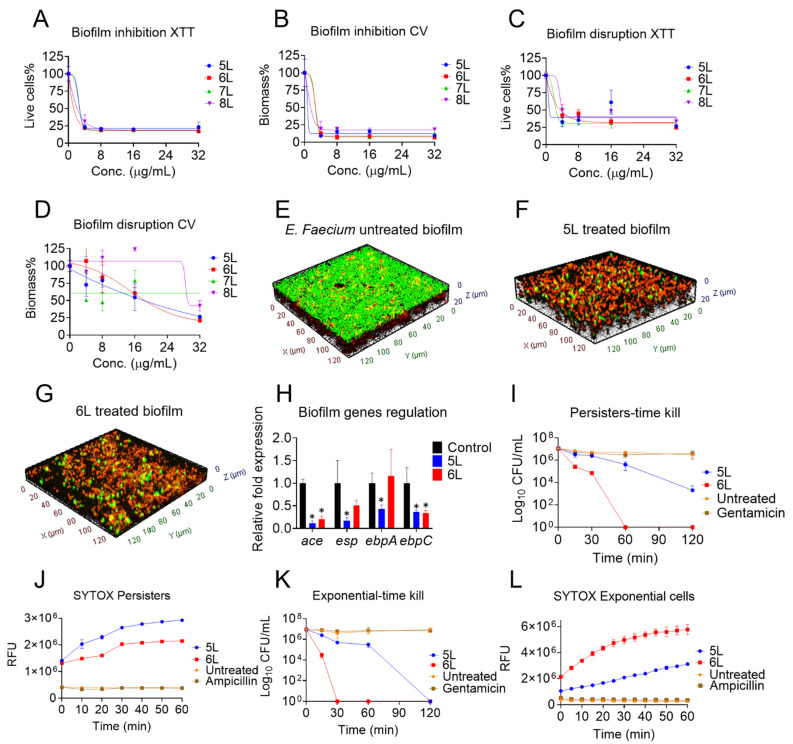
Antibiofilm and antipersister activity of designed LfcinB6 peptides against *E faecium* strain C68. (**A**) Inhibition of biofilm formation by peptides 5L–8L, live-cell reductions by XTT and (**B**) biomass by CV. (**C**) Disruption of 24 h established biofilms, live-cell reductions by XTT, and (**D**) biomass by CV. (**E**) Confocal microscopy of control *E. faecium* biofilm and (**F**) 5L- and (**G**) 6L-treated biofilms at 32 µg/mL. (**H**) Key biofilm genes regulation by peptides 5L and 6L via real-time PCR (significant, * *p* < 0.05, determined by Student’s *t*-test) (**I**) The kinetic killing of *E. faecium* strain C68 persister cells by 5L and 6L at 10× MIC. (**J**) SYTOX-based membrane permeabilization of persister cells by peptides 5L and 6L at 16 µg/mL. (**K**) The kinetic killing of *E. faecium* strain C68 exponential cells by 5L and 6L at 10× MIC. (**L**) SYTOX-based membrane permeabilization of exponential cells by peptides 5L and 6L at 16 µg/mL.

**Figure 2 antibiotics-11-01085-f002:**
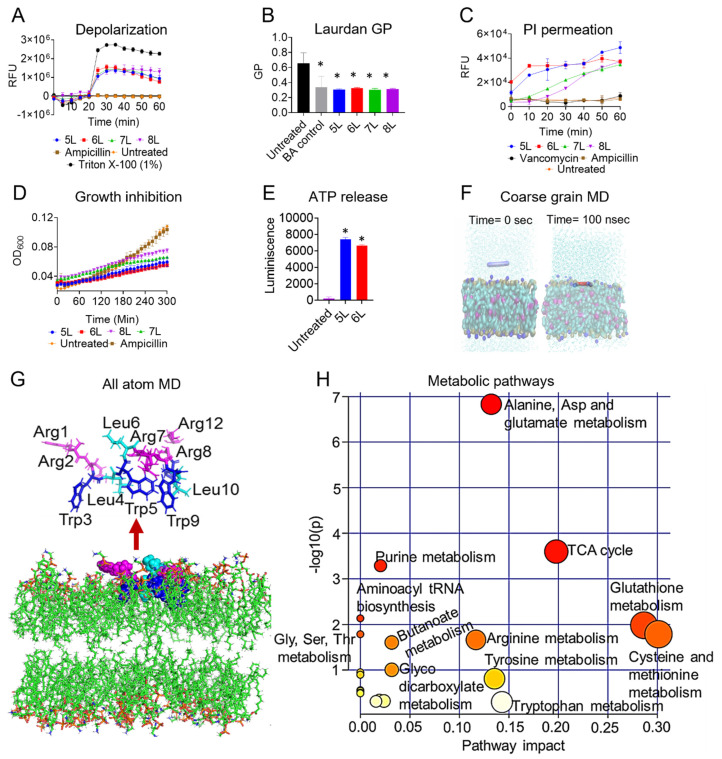
A plausible mechanism of action of LfcinB6-derived peptides. (**A**) DIBAC4(3) assisted *E. faecium* strain C68 membrane depolarization caused by peptides 5L–8L at 16 µg/mL. (**B**) Laurdan-based generalized polarization assessment by peptides at 32 µg/mL (significant, * *p* < 0.05, determined by one way ANOVA). (**C**) PI permeation and (**D**) growth inhibition by 5L–8L peptides at 32 µg/mL. (**E**) ATP leakage of *E. faecium* strain C68 by peptide 5L and 6L at 32 µg/mL (significant, * *p* < 0.05, determined by one way ANOVA). (**F**) Snapshot of coarse grain MD simulation of peptide 5L in the presence of POPC: cardiolipin (3:1) membrane mimetic model showing complete peptide insertion at 100 ns. (**G**) All atom dynamics of peptide 5L showing a non-helical and amphipathic distribution of amino acids (magenta: arginine; blue: tryptophan; and cyan: leucine). (**H**) Pathway analysis of the targeted metabolome of *E. faecium* strain C68 treated with peptide 5L at 2× MIC.

**Figure 3 antibiotics-11-01085-f003:**
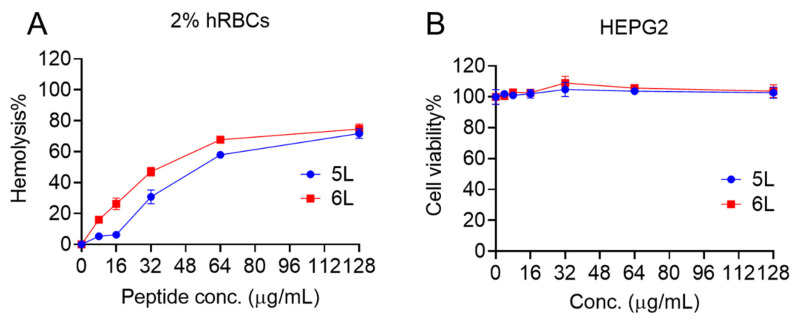
Cytotoxicity of LfcinB6-derived peptides. (**A**) Hemolysis potential of 5L and 6L to human red blood cells, (**B**) cellular toxicity to HepG2 cell lines.

**Figure 4 antibiotics-11-01085-f004:**
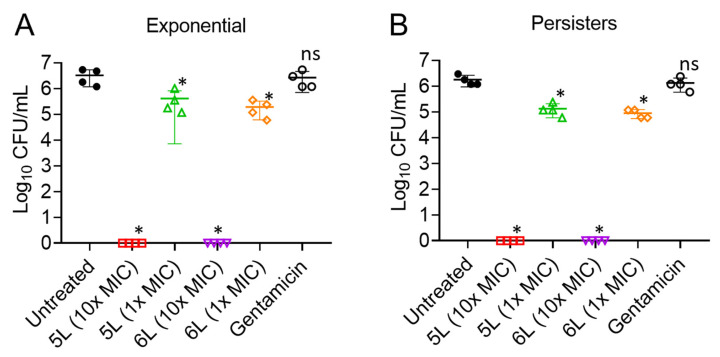
Efficacy of designed LfcinB6 peptides in an ex vivo porcine skin infection model infected with *E. faecium* C68. (**A**) Bacterial load quantitation (exponential cells) after treatment with peptides 5L and 6L at 10× or 1× MIC. (**B**) Bacterial load quantitation (persister cells) after treatment with peptides 5L and 6L at 10× or 1× MIC. (* *p* < 0.05 are significant, determined by one way ANOVA). ns: not significant.

**Table 1 antibiotics-11-01085-t001:** Physical parameters and the minimal inhibitory concentration (µg/mL) of designed peptides against *E. faecium*.

Peptide	Sequence ^a^	NC ^b^	Hph% ^c^	Hy ^d^	Hm ^e^	rT ^f^ (min)	MIC (µg/mL)
							EF C68 ^g^
1L	RRWQWR**LLLL**-NH_2_	3	60	0.805	0.168	18.083	16
2L	RRW**L**WR**L**-NH_2_	3	57	NP	NP	13.321	>32
3L	**L**RW**L**WRRRW**L**WR-NH_2_	5	58	0.754	0.146	16.145	16
4L	R**L**W**L**WRRRW**L**WR-NH_2_	5	58	0.754	0.219	18.377	4
5L	RRW**L**W**L**RRW**L**WR-NH_2_	5	58	0.754	0.333	17.522	4
6L	RRW**L**WR**L**RW**L**WR-NH_2_	5	58	0.754	0.425	18.603	4
7L	RRW**L**WRR**L**W**L**WR-NH_2_	5	58	0.754	0.214	16.290	4
8L	RRW**L**WRRRW**L**W**L**-NH_2_	5	58	0.754	0.075	21.235	4
Amp	N.P.	N.P.	N.P.	N.P.	N.P.	N.P.	>32
Van	N.P.	N.P.	N.P.	N.P.	N.P.	N.P.	>32

^a^ Peptide sequences have free N-terminus and amidated at C-terminus; ^b^ NC denotes net charge; ^c^ Hph% represents the hydrophobic amino acid compositions (total hydrophobic ratio) in the peptide; ^d^ Hy: Hydrophobicity and ^e^ Hm: hydrophobic moment of respective peptides calculated from HeliQuest analysis (https://heliquest.ipmc.cnrs.fr/, accessed on 20 June 2022); N.P. is Not predictable by the HeliQuest software; ^f^ HPLC retention time in mins on a C18 reverse-phase column; ^g^
*E. faecium* strain C68 is ampicillin (Amp) and vancomycin (Van) resistant [40].

**Table 2 antibiotics-11-01085-t002:** MIC (µg/mL) of designed peptides against *E. faecium* strain C68 in the presence of physiological salt concentration.

Peptide	MIC (µg/mL)				
	Media Only	+ NaCl (150 mM)	+ CaCl_2_ (2.5 mM)	+ ZnSO_4_ (8 µM)	+ MgSO_4_ (1 mM)
1L	16	4	32	4	8
2L	>32	N.A.	N.A.	N.A.	N.A.
3L	16	16	>32	4	8
4L	4	8	>32	4	8
5L	4	4	16	4	8
6L	4	4	8	4	8
7L	4	4	8	2	4
8L	4	4	16	2	4
Amp ^a^	>32	>32	>32	>32	>32

^a^ Ampicillin; N.A.: Not available.

**Table 3 antibiotics-11-01085-t003:** MIC (µg/mL) of selected bovine lactoferrin peptides against *E. faecium* clinical isolates.

Peptide	MIC (µg/mL)					
	D14	D24	D25	D29	E007	WC176
5L	16	8	4	16	4	8
6L	8	8	4	8	4	8
7L	8	4	4	8	4	16
8L	16	8	8	32	4	16
Ampicillin	≥32	>32	>32	>32	>32	>32
Vancomycin	2	1	1	1	1	16

## Data Availability

Data is contained within the article and Appendix A.

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
