# Peer review of "Design and Evaluation of Short Bovine Lactoferrin-Derived Antimicrobial Peptides against Multidrug-Resistant Enterococcus faecium"

_antibiotics, 2022, doi:10.3390/antibiotics11081085_

Round 1

Reviewer 1 Report

The paper by Mishra et al titled "Design and Evaluation of Short Bovine Lactoferrin-derived Antimicrobial Peptides against Multidrug-Resistant Enterococcus 3 faecium" is a good and well-written paper that describes the development of short LfcinB6 peptides with activity against E. faecium. After a few minor adjustments, I recommend it for publication. My only two minor observations are:
1. Please rectify the 96-94.
2. Please determine the resolution of figure 1 EFG.

Reviewer 2 Report

The authors have described the Bovine Lactoferrin-derived small peptide sequence as an antimicrobial agent. It has shown high potency against Multidrug-Resistant Enterococcus 3 faecium. They have established antimicrobial activity through various assays/studies such as membrane permeability, growth inhibition assay, ATP assay, and in vivo study. This is an excellent article for this journal. It should be published with a minor revision.

i)                     Motive behind the choosing LfcinB6 sequence as an antimicrobial activity should be given.

ii)                   In vivo study, the data should be compared with vancomycin or any existing antimicrobial drug.

Reviewer 3 Report

The authors designed and synthesized a series of AMPs, and found peptide 5L showed good potential as a new antibiotic candidate against E. faecium. The research work is comprehensive and results are sound and solid. The article fits the aim and scope of this journal. We only have several minor suggestions.

1.     Line 63. “Their attractive toxicity profile make them excellent drug candidates”. This sentence is ambiguous. The relationships between toxicity and drug application should be explained in detail.

2.     Line 88. For better understanding, the strains of these bacteria should be list here instead of list in the supplementary file.

3.     Line 92-93, 314. The font size should be adjusted.

4.     Line 256. The unit A° should be Å.

5.     The peptide design concept was not clear enough in the current form. It can be described more detailed, and move it to the method part.

6.     Figure 1 E-G, the resolution should be improved.

7.     The peptides were predicted to be α-helical. We suggest it could be experimentally validated by using circular dichroism or other methods. Or cite your previous work which can prove the α-helical property of these peptides.

8.     Why the authors chose the application assay by using ex vivo porcine skin infection model instead of alive murine skin infection model?

9.     The resolution of Table S5-6 should be improved. Please provide tables instead of pics.

Reviewer 4 Report

The manuscript submitted to Antibiotics journal investigates the anti-bacterial activity of new short bovine lactoferrin-derived antimicrobial peptides against the multidrug-resistant Enterococcus faecium. The subject of the work is interesting and important due to the potential use of these designed peptides in bacterial control. However, there are some gaps and shortcomings in the manuscript, which must be corrected before considering its publication. Detailed comments for consideration are provided below:

#Abstract and introduction sections: The authors did not convince us how the “L” residue is important to the short AMPs design and what can be improved with the “R” replacing with “L” residues. Even though the last paragraph of introduction mentioned that “We designed a series of peptides with either one or two linear copies of RRWQWR motifs with aided hydrophobic increments and selective amino acid permutations that was specifically active against E. faecium.”, the case studies/evidences or the hydrophobic increments by replacing “R” with “L” amino acids should be mentioned right before this paragraph to engage the readers.

 #Materials&methods section:

Page 2 line 88, “Bacterial strains of E. faecium, E. faecalis, and Staphylococcus aureus…..”, you did not mentioned about S. aureus before and I am quite sure that it’s not “a strain” of E. faecium. What is the purpose of having S. aureus here? Please be more specific since it was not included in the objective(s) of this study. If it is for comparison, it should be clearly declared at this step.

page 2 line 91-94, is there any specific reason for the larger font size in some phrases here?

Page 3 line 100-102, I believe that the MIC cannot be “measured” by spectrophotometer. You can measure the absorbance at OD600 to determine the bacterial growth and based on that you can determine the MIC at specific concentration. Please rewrite this part.

Page 3 line 104-106, You have never mentioned the role of “various salt concentrations” as a “independent variable” that affect to the activity of peptides before. If this variable is so important, I recommend you to mention about this issue in the “introduction” part and also provide some evidences with citations. So, in this materials&methods you can specifically use the sub heading as “Minimal inhibitory concentration (MIC) assayof designed peptides in various salt concentrations”.

#Results:

Figure 1 E,F,G: the resolution were poor. The sub pictures were unclear and the axes scale were unreadable. The orientation angle of the detected biofilms should be in the same viewing angles. Please specify the “control” either it is “untreated” condition or treated with the negative agent.

Figure 2 F,G,H: the sub pictures were too small. They can either be expanded or isolated as figure 3, 4, 5. The information from Fig.2H is really important but the text was too small, I think it deserved to be shown at larger size.

#Dicussions:

Page 14 line 543-552, the roles and effects of salt to the peptides were mentioned but lack of specific evidences of particular salt types (as you tested in NaCl, CaCl2, ZnSO4, and MgSO4). It would be nice if you can interpret the effects of “various salt concentrations” as you proposed.

Page 14 line 579-80, “arg residues” should be “Arg residues”, “trp” should be “Trp” And then again, the point of replacing “R” with “L” residues should be discussed here in this paragraph with enough of supporting evidences.
